# Overexpression of Bacterial Beta-Ketothiolase Improves Flax (*Linum usitatissimum* L.) Retting and Changes the Fibre Properties

**DOI:** 10.3390/metabo13030437

**Published:** 2023-03-17

**Authors:** Justyna Mierziak, Wioleta Wojtasik, Anna Kulma, Magdalena Żuk, Magdalena Grajzer, Aleksandra Boba, Lucyna Dymińska, Jerzy Hanuza, Jakub Szperlik, Jan Szopa

**Affiliations:** 1Department of Genetic Biochemistry, Faculty of Biotechnology, Wroclaw University, Przybyszewskiego Str. 63, 51-148 Wroclaw, Poland; 2Department of Dietetics and Bromatology, Faculty of Pharmacy, Wroclaw Medical University, Borowska 211, 50-556 Wroclaw, Poland; 3Department of Genetics, Plant Breeding and Seed Science, Wroclaw University of Environmental and Life Sciences, Grunwaldzki Sq. 24A, 50-363 Wroclaw, Poland; 4Department of Bioorganic Chemistry, Wroclaw University of Economics and Business, Komandorska 118/120, 53-345 Wroclaw, Poland; 5Institute of Low Temperature and Structure Research, Polish Academy of Sciences, Okólna 2, 50-422 Wroclaw, Poland; 6Laboratory of Tissue Culture, Botanical Garden, Faculty of Biological Sciences, University of Wroclaw, Sienkiewicza 23, 50-525 Wroclaw, Poland

**Keywords:** beta-ketothiolase, 3-hydroxybutyrate, polyhydroxybutyrate, flax fibre, flax stem, cell wall

## Abstract

Beta-ketothiolases are involved in the beta-oxidation of fatty acids and the metabolism of hormones, benzenoids, and hydroxybutyrate. The expression of bacterial beta-ketothiolase in flax (*Linum usitatissimum* L.) results in an increase in endogenous beta-ketothiolase mRNA levels and beta-hydroxybutyrate content. In the present work, the effect of overexpression of beta-ketothiolase on retting and stem and fibre composition of flax plants is presented. The content of the components was evaluated by high-performance liquid chromatography, gas chromatography–mass spectrometry, Fourier-transform infrared spectroscopy, and biochemical methods. Changes in the stem cell walls, especially in the lower lignin and pectin content, resulted in more efficient retting. The overexpression of beta-ketothiolase reduced the fatty acid and carotenoid contents in flax and affected the distribution of phenolic compounds between free and cell wall-bound components. The obtained fibres were characterized by a slightly lower content of phenolic compounds and changes in the composition of the cell wall. Based on the IR analysis, we concluded that the production of hydroxybutyrate reduced the cellulose crystallinity and led to the formation of shorter but more flexible cellulose chains, while not changing the content of the cell wall components. We speculate that the changes in chemical composition of the stems and fibres are the result of the regulatory properties of hydroxybutyrate. This provides us with a novel way to influence metabolic composition in agriculturally important crops.

## 1. Introduction

Flax (*Linum usitatissimum* L.) is a well-known and valued cultivated plant, and its products—fibres and oil—are used by many industries. Flax fibres are obtained from straw. Traditionally, flax fibres are used in the production of textiles, but due to the content of phenylpropanoid and terpenoid compounds with pro-health properties, it can be used for the production of wound healing dressings, biodegradable packaging, and implants. It can also serve as a raw material for the production of composite materials. The whole flax straw can be used for the production of biofuels, animal bedding, non-structural building materials, specialty papers, and as an source of components for alternative cancer therapies [1,2,3,4,5]. The chemical composition of straw is mainly cellulose, hemicellulose, pectin, and lignin. These polymers are present in the primary and secondary cell wall, which are responsible for structural support. The cell wall is a protective barrier and plays an important role in the plant response to pathogenic infections and other environmental stresses. It is also important during the various stages of development [6]. Due to the versatile use of flax raw materials, there is a constant effort to improve the utility properties of this crop.

Beta-ketothiolases have many important functions in plant cells. The substrate for beta-ketothiolases is acetyl-CoA, a key compound at the crossroads of the anabolic and catabolic processes in plant cells. Acetyl-CoA is involved in the synthesis of fatty acids, amino acids, glucosinolates, as well as other secondary metabolites, including flavonoids, stilbenes, and isoprenoids. Acetyl-CoA is also used for the acetylation of secondary and primary metabolites [7]. Beta-ketothiolases are involved in the beta-oxidation of fatty acids, which is an important process during germination as it degrades the fatty acids stored in the seeds. Beta-oxidation is also a key process that determines the lipid composition of plants. In addition, this cycle plays an important role in the development of reproductive tissues and seeds. Studies on the *Arabidopsis thaliana* mutant indicate that thiolases not only play a role in plant reproduction, including germination and seedling growth, but disturbances in the processes carried out by thiolases also lead to abnormalities in vegetative growth [8,9,10]. Plants with double silencing of the beta-ketothiolase genes (KAT2 and KAT5) were infertile and showed severe morphological defects in both vegetative and reproductive organs [11]. It has been shown that thiolases are also involved in plant aging processes [12]. Beta-ketothiolases are also involved in the biosynthesis of plant hormones such as jasmonic acid (JA) and indole-3-acetic acid (IAA) and thus play an important role in hormonally regulated processes [9,13,14,15,16]. Studies conducted on *Arabidopsis thaliana* showed the involvement of beta-ketothiolase 2 (KAT2) in wound-induced jasmonic acid biosynthesis [17,18]. KAT2 overexpression increases jasmonic acid (JA) biosynthesis and accelerates dark-induced leaf senescence [19] and inhibits flower development controlled by IAA [9]. In addition, KAT2, by modulating ROS homeostasis, can regulate ABA signalling in processes such as ABA inhibition of seed germination and seedling growth, and stomatal closure and opening [19]. Beta-ketothiolases are also involved in the biosynthesis pathway of benzenoid compounds that perform many important functions in the plants such as growth regulation, protection against environmental stress, and defensive signalling. Changes in the expression of beta-ketothiolases affect the content of benzenoids in plants [20,21,22]. Studies on the effect of activation of the phenylpropanoid pathway in flax during infection with the pathogen *Fusarium oxysporum* showed an increase in the expression of the gene encoding beta-ketothiolase [23]. In addition, literature data indicate that the thiolase involved in the metabolism of terpenoids is an enzyme involved in the adaptation of plants to abiotic stress, such as salinity and low temperature [24]. One of the reactions carried out by beta-ketothiolase is the condensation of two acetyl-CoA molecules with subsequent reduction leading to the synthesis of 3-hydroxybutyrate (3-HB) in animals and its polymer polyhydroxybutyrate (PHB) in bacteria. The 3-HB in vertebrates can serve as a metabolic fuel in tissues. It can originate from direct synthesis from acetyl-CoA and probably additionally from hydrolysis of gut microbiota-derived PHB. In animals, 3-HB synthesis is increased during metabolic stress conditions. 3-HB is known for inhibition of histone deacetylases, and subsequent influence on gene expression [25,26,27]. The presence of 3-HB in plants was discovered very recently. It was identified for the first time in transgenic and non-transgenic flax, and showed the potential to regulate chromatin remodelling and gene expression in plants grown in vitro [28].

Previously the presence of PHB in plants was only reported as the result of the expression of three bacterial genes (including beta-ketothiolase) in transgenic plants [29], including flax [30]. In flax, besides accumulation of PHB, various metabolic changes were observed. In these plants, the content of fatty acids, sugars, and phenylpropanoid compounds also changed and they showed increased resistance to fungal infections [30,31,32]. The retting of the straw in those plants was also more efficient. The increase in retting efficiency could have been influenced by the decreased level of pectin in the flax straw. The structure of the fibre itself was also changed. The raw materials from transgenic plants with overproduction of PHB also exhibit a number of biomedical properties [33,34,35,36]. It can be speculated that the observed metabolic changes in flax overproducing PHB, as well as its pro-health properties, could be caused by the activity of 3-HB. The results of research by Tsuda et al. suggest that PHB may also be endogenously produced in plants [37].

The aim of the study was to investigate whether overexpression of only one gene—beta-ketothiolase—affects the retting process of flax straw and the quality of the flax fibres. 3-HB as a regulatory molecule performs important functions during plant development, and its polymer is built into the cell wall, so it was important for us to determine the content of these compounds in the stems of mature plants. In addition, before the retting process, we determined the composition of the cell wall and phenolic and terpene compounds. On the other hand, the quality of the flax fibres was determined mainly on the basis of the structure and the content of polymers in the cell wall and the content of phenolic compounds.

## 2. Materials and Methods

### 2.1. Research Material

The flax variety used in experiments (*Linum usitatissimum* L. cv. NIKE) was obtained from the Flax and Hemp Collection of the Institute of Natural Fibres and Medicinal Plants (Poznań, Poland). The generation of a gene construct and transgenic flax overexpressing beta-ketothiolase (line C10 and line C47) have been previously described [30]. To confirm the presence of the transgene in of the flax, the mRNA level of the introduced bacterial beta-ketothiolase (phbA) gene and endogenous beta-ketothiolase and 3-HB contents were determined (Appendix A).

The 3rd generation flax seeds were sown in a greenhouse. Greenhouse growth conditions: breeding medium: peat soil (pH 7.0) with sand (2:1); moisture: 60%/16 h, 75% 8 h; temperature: 22 °C/16 h, 15 °C 8 h; illumination: 103 mmol/s/m2 (50%), 250 mmol/s/m^2^ (100%), day/night 16/8 h. The plants were watered daily. Some of 5-week-old plants were collected for further analysis and the rest were cultivated until maturity. At the last stage, the flax was collected and dried, and then separated into stems, roots, and seeds. After drying the flax stems, scutching and heckling of the fibres was performed.

### 2.2. Isolation of Total RNA

The RNA was isolated with TRIzol reagent (Thermo Fisher Scientific, Waltham, MA, USA) using the manufacturer’s procedure. To eliminate residual genomic DNA, digestion was carried out with DNase I (Thermo Fisher Scientific, Waltham, MA, USA).

### 2.3. cDNA Synthesis on RNA Template (Reverse Transcription)

The cDNA was synthesized using the High Capacity cDNA Reverse Transcription Kit (Thermo Fisher Scientific, Waltham, MA, USA) using the manufacturer’s protocol.

### 2.4. Determination of mRNA Level

The mRNA level was determined by the real-time PCR method as previously described using the primers listed in Appendix A in [28].

### 2.5. Determination of 3-HB and PHB in Plant Tissues

The measurement of 3-HB and PHB levels were conducted according to Bohmert et al. with some modifications. Approximately 50–100 mg of plant tissues was weighed into a glass tube and then was incubated with 1 mL of methanol for 1 h at 55 °C to remove the lipids. The tissues were extracted twice with ethanol for 1 h at 55 °C to collect unpolymerized 3-hydroxybutyrate monomers. Then, 1 mL of chloroform and 1 µL of the internal standard 3-hydroxy-valerate-methyl ester were added to the remaining tissue, following by 1.7 mL ethanol and 0.2 mL HCL and incubation for 4 h at 100 °C. To separate the chloroform layer from the ethanol one, 4 mL of 0.9 M NaCl was added. The chloroform layer was collected into 4 mL vials containing sodium sulphate and incubated until the next day in order to absorb any remaining water. To 25 µL of the extract, 75 µL of n-methyl n-trimethylsilyltrifluoroacetamide was added and then heated for 45 min at 37 °C. An aliquot (1 µL) of the derivatized mixture was injected into a GC-MS/MS system (PAL RSI 85, GC Agilent Technologies 7890B, and a 7000D GC/TQ triplequad mass spectrometer) using a split of 1:50. The inlet was set at 250 °C; helium flow set at 0.8 mL/min; oven temperature was 80 °C: hold for 2 min, ramp 10 °C/min up to 245 °C and hold for another 2 min; and total run time 20.5 min. The GC was equipped with a ZB-5MS 30 m × 0.25 mm × 0.25 µm (Phenomenex, Torrance, CA, USA). MRM mode was used to monitor the ion transition for ethyl 3-hydroxybutyrate, TMS derivative: 161–118.9 (quantitative ion for 3-HB) and 189–175 (quantitative ion for PHB), for ISTD 175–133.1 (quantitative ion) [38].

### 2.6. Determination of Phenolic Compound Content by UPLC Method

The content of phenolic compounds was determined using UPLC with PDA and MS detection as previously described in [39]. A total of 200 mg of ground flax tissue from mature stems and flax fibres was used for extraction.

### 2.7. Determination of Cellulose Content

The cellulose content was determined using an anthrone reagent after prior hydrolysis with sulfuric acid, calculated as the equivalent of a hydrolysed cellulose standard, as previously described [40]. A total of 15 mg of ground flax stem and fibres was used for the analysis.

### 2.8. Determination of Lignin Content

The lignin content was determined by the acetyl bromide method, calculated on the basis of a standard curve prepared using coniferyl alcohol, as previously described [41]. A total of 15 mg of ground flax stem and fibres was used for the analysis.

### 2.9. Fractionation of Cell Wall

The cell wall fractionation into a water-soluble fraction (WSF), CDTA-soluble fraction (CSF), Na_2_CO_3_ and NaBH_4_ soluble fraction (NSF), 1M KOH-soluble fraction (K1SF), and 4M KOH-soluble fraction (K4SF) was carried out according to the previously described method in [42].

### 2.10. Determination of Uronic Acids

The content of uronic acids was determined by a previously described method [43,44]. A total of 10 mg of lyophilized cell wall fractions was used for the analysis. The content of uronic acids was determined on the basis of spectrophotometric measurement at 525 nm. A standard curve was prepared using glucuronic acid.

### 2.11. Determination of Monosaccharides by UPLC Method

The content of monosaccharides was determined after derivatization by the UPLC method with PDA detection that was previously described [42]. A total of 10 mg of ground flax stem and fibres was used for the analysis.

### 2.12. Determination of Total Pectin and Hemicellulose

The total pectin content was calculated as the sum of uronic acids (determined spectrophotometrically by the biphenyl method) and monosaccharides (determined by the UPLC method) of the WSF, CSF, and NSF fractions. The total hemicellulose content was calculated as the sum of uronic acids and monosaccharides of the K1SF and K4SF fractions.

### 2.13. In Vitro Retting Experiment

Harvested flax stems were cut into 5 cm pieces and immersed in tap water. The samples were placed in open containers and incubated at 22 °C for 41 days, with weekly observation and photographic documentation.

### 2.14. Analysis of Fatty Acids by GC-FID Method

The analysis of fatty acids in flax stem using the GC-FID method was based on the conversion of fatty acids into their methyl esters as previously described [39].

### 2.15. Analysis of Terpenoid Compounds by UPLC Method

The content of terpenoids was determined by UPLC with PDA detection as previously described [45]. A total of 300 mg of lyophilized and ground flax stem were used for the analysis. After extraction, the pellet was suspended in 1 mL of acetonitrile:methanol (1:1). The integration of the peaks was carried out at 450 nm. The concentration of the compounds was determined based on the peak area (in triplicate) relative to the relevant carotenoid, tocopherol, and chlorophyll standards.

### 2.16. Analysis of Sterols by GC-FID Method

The content of sterols was determined by a previously described method [45]. A total of 200 mg of ground flax stem was used for the analysis

### 2.17. IR and X-ray Analysis

IR spectra were measured using a Nicolet iS50 FT-IR (Thermo Scientific) spectrometer equipped with an Automated Beamsplitter exchange system (iS50 ABX containing a DLaTGS KBr detector and DLaTGS Solid Substrate detector for mid-IR and far-IR regions, respectively). A built-in all-reflective diamond ATR module (iS50 ATR), Thermo Scientific Polaris™, and HeNe laser were used as an IR radiation source. Polycrystalline mid-IR spectra were collected in the 4000–400 cm^−1^ range in KBr pellets and Nujol mulls. Spectral resolution was set to 4 cm^−1^. In the quantitative analysis of the measured IR spectra, the ORIGIN program was used for deconvolution of the complex spectral contours into Lorentzian components.

X-ray powder diffraction patterns were recorded at room temperature using X’Pert PRO powder diffractometer (PANalytical, Almelo, The Netherlands) working in the transmission or reflection geometry, equipped with a linear PIXcel detector and using CuK_α1_ radiation (λ = 1.54056 Å) in the 2θ range from 5° to 100° with a step of 0.03°.

### 2.18. Determination of the Antioxidant Potential of Flax Stem Extract by Spectrophotometric Method Using DPPH

Stem extracts obtained in accordance with the procedure described in Section 2.6 were used to determine the antioxidant potential. A volume of 1 mL of DPPH (2,2-diphenyl-1-picryl-hydrazyl-hydrate) reagent in methanol was added to each Eppendorf tube followed by 30 µL of stem extract. Incubation was carried out for 15 min in the dark, and then the absorbance was measured at 515 nm. The control sample was 1 mL of DPPH and 30 µL of methanol. The blank sample was methanol.

### 2.19. Statistical Analysis

The results are expressed as the mean of three biological repetitions ± standard deviation (SD) and statistically different changes were calculated using ANOVA (with Fisher post hoc) with use of Statistica software (Staltsoft, Tulsa, OK, USA).

## 3. Results

### 3.1. Productivity of Transgenic Flax

Transgenic plants with overexpression of the bacterial beta-ketothiolase gene showed no phenotypic changes. No differences in shape, height, or colour and size of leaves or flower petals were observed between transgenic and control plants. The flowering time was the same for transgenic and non-transgenic plants. The productivity of the obtained plants was assessed by analysing the two main flax raw materials, seeds and stems (Appendix A). The average length of the flax stem did not differ. The weight of the seeds was slightly lower and thus the weight of the seed baskets was also slightly lower. However, the average number of seeds in the seed basket did not differ, so the number of germinated plants from seeds obtained from the same cultivated area did not deteriorate.

### 3.2. Determination of 3-Hydroxybutyrate and Polyhydroxybutyrate Content

The bacterial beta-ketothiolase gene introduced into plants encodes an enzyme involved in the synthesis pathway of 3-hydroxybutyrate and its polymer (PHB) in bacteria. Our previous research has shown that 3-HB is a compound that is also found in wild type flax [28]. The content of 3-HB and PHB was determined at the early vegetation of flax development (5-week-old plants) and in the mature plants (flax stem) (Figure 1). The content of 3-HB in the 5-week-old non-transgenic plants was 0.0881 µg/g and in the flax stem was 1.58 µg/g, while the content of PHB in the 5-week-old non-transgenic flax was 0.261 µg/g and in flax stem was 0.185 µg/g. The transgenic flax line C contained approximately 1.5 times more 3-hydroxybutyrate at the early vegetation stage compared with non-transgenic plants (0.142 µg/g in line C47 and 0.134 µg/g in line C10). The content of PHB was reduced in transgenic plants at this stage of development and was 0.2 µg/g in the C10 line and 0.079 µg/g in the C47 line. A decrease in 3-HB content was observed in the mature stems of flax. The largest decrease was recorded for line C47. This line contained an average of 0.71 µg/g and it was 55.4% less than that in non-transgenic plants. This transgenic line also showed the highest decrease in PHB content in straw (19.5% decrease compared to control). The transgenic line C10 contained 1.23 µg/g of 3-hydroxybutyrate, 19% less compared to non-transgenic plants; however, in flax stems, the PHB content was 52% higher than in the control.

In summary, the presence of free beta-hydroxybutyrate in flax was demonstrated, and its content varied depending on the stage of flax development. Transgenic plants were characterized by an increased content of 3-HB at the early vegetation stage (5-week-old plants) and reduced content at the stage of flax maturation (stems) compared with non-transgenic plants. The content of polyhydroxybutyrate was reduced in the transgenic flax, except for the stems of C10 plants, which showed a significant increase in this compound.

### 3.3. Content of Phenolic Compounds in Flax Stems

The content of free phenolic compounds (Figure 2a) and those bound to the cell wall (Figure 2b) was determined in the transgenic flax stems. Of the free phenolic compounds, we identified two benzene-based compounds (vanillin and vanillic acid) and four flavonoids (6,8-C-diglucoside of apigenin, 6-C-glucose of luteolin, 8-C-glucoside of luteolin, and 6-C-glucoside of apigenin). In the stems of transgenic flax, vanillin, vanillic acid, syringaldehyde, luteolin 8-C-glucoside, apigenin 6-C-glucoside, p-coumaric acid, caffeic acid, and ferulic acid were identified as compounds bound to the cell wall.

In transgenic flax lines, a lower level of both free vanillic acid (80% reduction in line C47 and 30% in line C10) as well as vanillic acid bound to the cell wall (22.7% reduction in line C10 and 27.2% in line C47) was observed when compared with non-transgenic flax. The content of free vanillin was significantly lower only in line C47 (by 52.7%). The content of vanillin bound to the cell wall was two times lower in line C10 and was 19.8% lower in line C47 than in the control plants. Line C47 contained much more free flavonoids—6-C-glucoside of apigenin, 6,8-C-diglucoside of apigenin, 8-C-glucoside of luteolin, and 6-C-glucoside of luteolin—than control plants. Line C10 also showed a higher 6,8-C-diglucoside of apigenin and 6-C-glucoside of luteolin content.

For flavonoids associated with the cell wall, a lower content of 6-C-glucoside of apigenin for line C10 and 8-C-glucoside of luteolin for line C47 were observed. A lower content associated with the cell wall of caffeic acid was observed for both lines. The content of p-coumaric acid for line C47 was two times lower than that in the control, while line C10 showed a 1.7-fold higher value. Line C10 showed significantly lower syringic aldehyde content, while line C47 showed a similar level of this compound in flax stems compared with the control line.

### 3.4. Content of Carotenoids and Chlorophylls in Flax Stems

The content of all measured terpenoid compounds was lower in the mature stems of transgenic flax lines C10 and C47 compared with the non-transgenic flax (Figure 3). Line C10 contained half of the chlorophyll a and chlorophyll b content of the control plants. Line C47 showed a 13.8% lower chlorophyll b content compared with non-transgenic plants, while the content of chlorophyll a did not differ. Both transgenic plant lines contained about 32% less beta-carotene compared with non-transgenic flax stems. The largest difference was observed in the case of lutein. Line C47 contained 80% less of this compound compared with the control plants, while C10 line showed 90% less. The last tested compound from the carotenoid group was violaxanthin. The C47 line contained slightly less of this compound than control plants, while the C10 line contained 57.8% less.

### 3.5. Content of Phytosterols in Flax Stems and Seeds

The content of phytosterols (campesterol, beta-sitosterol, and stigmasterol) in the stems and seeds of transgenic flax lines C10 and C47 is presented in Appendix A, respectively. The content of each phytosterol, in both the stems and seeds, in the transgenic flax lines did not differ significantly from non-transgenic flax.

### 3.6. Content of Fatty Acids in Flax Stems and Seeds

Due to the beneficial properties of flax fatty acids for human health and the fact that beta-ketothiolase is an important enzyme involved in beta-oxidation of fatty acids, their content was determined in the stems (Figure 4) and seeds (Appendix A) of transgenic flax lines C10 and C47. The total amount of fatty acids (palmitic acid; C16:0, stearic acid; C18:0, alpha-linolenic acid; C18:3 alpha, linoleic acid; C18:2, oleic acid; and C18:1) in both the stem and seeds of both transgenic flax lines was lower than that in control plants.

The stems of line C10 contained 32.0% less fatty acids than the stems of non-transgenic plants, while those of line C47 contained 23.3% less. The seeds of line C10 had 17.5% less, and line C47 had 10.2% less fatty acids than non-transgenic flax seeds. The content of individual fatty acids in transgenic flax line C47 stems was reduced except for palmitic acid. The largest difference was recorded for oleic acid: line C10 contained 68.9% less, and line C47 had 52.5% less compared with the control. In seeds, the largest difference was recorded for linoleic acid. The seeds of transgenic line C10 contained 28.2% less, and line C47 had 26.7% less than non-transgenic flax seeds. The amount of palmitic and oleic acid did not differ between the seeds of the transgenic and control flax lines.

### 3.7. Content of Cell Wall Components in Flax Stem

The content of polymers of the cell walls of transgenic and non-transgenic flax stems is shown in Figure 5 and Figure 6. Cellulose is the main building polymer of plant cell walls, while lignin is the key polymer responsible for the strength of the stem, thus allowing its growth and vertical orientation. The content of cellulose and lignin in flax stems did not show statistically significant differences between transgenic flax lines with overexpression of beta-ketothiolase and control flax. Only line C47 had significantly less lignin compared with the control flax. The cellulose content in line C10 was about 0.328 mg/g and so its content was 5% higher compared to the control plants. On the other hand line C47 showed 9% lower cellulose content compared with non-transgenic flax. This line also contained 19.6% less lignin compared with the control. Line C10 also showed less lignin, but the difference was not as significant (4.6%). The ratio of lignin to cellulose is an important parameter indicating the mechanical properties of the plant cell wall. The higher the ratio of lignin to cellulose is the more rigid is the cell wall structure. This parameter was lower for the transgenic flax stems compared to those of the control flax.

In addition to cellulose, hemicellulose performs an important structural function, which is the cross-linking of cell wall components. The hemicellulose content in mature stems of transgenic flax line C10 did not significantly differ compared with control flax stems. No significant differences in the level of uronic acids included in the hemicellulose fractions were observed. Only a 20% higher monosaccharide level was observed. In addition, no changes were observed in the percentage of uronic acids and monosaccharides in the hemicellulose fractions. In contrast, transgenic line C47 had 13.4% lower monosaccharide content in the hemicellulose fractions and 17.3% lower total hemicellulose content. In the C47 transgenic line, the percentage of monosaccharides in the hemicellulose fractions changed significantly: there was an increase in the percentage of monosaccharides in the K1SF fraction, and thus a decrease in the K4SF fraction. However, no changes were observed in the percentage of uronic acids in the hemicellulose fractions.

The last polymer of the cell wall whose content was determined in mature stems was pectin. A slightly lower content of total pectin was observed in the stems derived from the transgenic flax lines, with only the changes in line C47 being statistically significant. This line contained 30% less pectin than the stems of non-transgenic flax. This difference was mainly due to a lower content of monosaccharide fractions in the stems of transgenic flax line C47. The analysis of total monosaccharides showed that the transgenic C47 line had 41% less of them compared to non-transgenic flax, while in the transgenic C10 line their amount did not change. On the other hand, differences in the percentage of monosaccharides in the pectin fractions were shown only in the C10 line, where more pectin in the WSF fraction and less pectin in the CSF fraction was observed. There were no differences between the transgenic lines and the control in the content of uronic acids in pectin. Only the C47 transgenic line was distinguished by a change in the percentage of uronic acids in the pectin fractions, which was more in the CSF fraction and less in the WSF.

### 3.8. Retting of Flax Stem

The retting of transgenic flax lines C10 and C47 as well as non-transgenic flax was carried out in laboratory conditions (Figure 7). The separation of fibres from woody parts of the stem, both in the transgenic lines and non-transgenic plants, was observed 21 days after the start of retting. In the transgenic flax lines, more separated fibres could be seen. This was confirmed by observations at subsequent time points, especially for transgenic line C47. Observations made at the last time point, i.e., 42 days after the start of retting, showed a significantly higher number of separated fibres in both transgenic flax lines compared with the control flax. To sum up, during retting in laboratory conditions in transgenic flax stems, no clear reduction in start-up time was observed, but the whole process was more efficient than in the case of control plants.

### 3.9. Content of Phenolic Compounds in Flax Fibres

The content of phenylpropanoid compounds was also determined in flax fibres. The fibres of the C10 line showed no significant changes in the content of free phenylpropanoid compounds compared to the fibres of the control plants, while the fibres of the C47 line contained significantly lower levels of these compounds (Figure 8a). The greatest decrease was noted for chlorogenic acid; the C47 line contained 94.3% less compared to the control. In the case of vanillic acid, which is a benzenoid, the C47 line showed a 69.0% lower content. In addition, in the C47 line, compared to non-transgenic plants, an almost two-fold decrease in the content of flavonoids was observed which included apigenin 6,8-diglucoside, apigenin 6-C-glucoside, luteolin 8-C-glucoside, and luteolin 6-C-glucoside. In the extract after alkaline hydrolysis of the cell walls in the flax fibres, three phenolic acids, one representative of benzenoids and one representative of flavonoids were identified (Figure 8b). The content of p-coumaric acid and ferulic acid bound to the cell wall in transgenic plants did not change compared to the control. The content of caffeic acid was significantly lower in each of the examined transgenic flax lines: the C10 line contained 48.8% less of this acid, and the C47 line had 75.7% less compared to that in the non-transgenic line. The content of flavonoids (apigenin 8-C-glucoside) did not change in the fibres of the transgenic plants relative to the control fibres. The content of vanillin, a representative of benzenoids, had the most significant decrease in the C10 line, which contained 55.3% less of this compound.

### 3.10. Content of Cell Wall Components in Flax Fibres

The analysis of the composition of the main polymers of the plant cell walls in the fibres showed no significant differences in the content of cellulose and lignin in the transgenic flax lines compared to control flax (Figure 9).

In the transgenic fibres of the C47 line, an increase of 79.4% of hemicelluloses was observed compared to their content in control plants (Figure 9). This increase resulted from the increase in the content of monosaccharides in this line (Figure 10). The content of sugars in both hemicellulose fractions changed.

All transgenic plants showed changes in pectin content in flax fibres compared to control (Figure 9). The C10 line contained 26.5% more pectin, while the C47 line showed a 20.1% decrease in their content in the fibres. The transgenic lines were characterized by changes in the content of simple sugars included in pectin (Figure 10). Compared to the control, Line C10 contained 47.3% more simple sugars and line C47 had 31.0% more in their fibres. The changes in the content of simple sugars occurred mainly in the fraction of pectin loosely bound to the cell wall (WSF). The total amount of uronic acids in all pectin fractions of the fibres did not change significantly compared to their content in the control fibres. In the transgenic fibres, however, there was a noticeable decrease in one of the fractions, the WSF fraction, in favour of the other two fractions.

### 3.11. The IR Analysis of Flax Fibres

Vibrational spectroscopy is widely used in the studies of bioorganic and natural systems. It is particularly suitable in the studies of genetically modified organisms, allowing the discrimination between them and non-transgenic organisms. Figure 11a presents their IR spectra recorded in the range of 400–4000 cm^−1^. The characteristic differences of the spectral contours were observed in the ranges 4000–2500, 1750–1480, 1480–1200, 1200–950, and 940–860 cm^−1^ which may be used for comparison of the cellulose, pectin, and lignin contents in the studied samples.

After quantitative analysis of the spectra, the integral intensity of the strong band at 2920 cm^−1^ was chosen as a standard (area under spectral band) for comparison with the intensity of other IR bands. This band corresponds to the νas(CH3) vibrations according to the usual procedure [46]. This band appeared on the slope of a very strong contour observed in the broad range of 2000–3750 cm^−1^ with the maximum at 3400 cm^−1^. It originates from the stretching vibrations of the free OH group and the one engaged in the inter- and intra-molecular interactions. Although the shape of this contour was similar for all studied samples, their deconvolution into Lorentz components allows the visualization of the essential differences between them. This can be seen from the comparison of the peak positions and integral intensities of the respective Lorenzian components presented in Figure 11b.

The band at about 3500 cm^−1^ corresponds to the stretching ν(OH) vibration of the free hydroxyl groups. The component in the 3455–3410 cm^−1^ range should be assigned to the intra-molecular hydrogen bond 2-OH···O-6 of cellulose. On the other hand, the band in the 3375–3340 cm^−1^ range corresponds to the 3-OH···O-5 intra-molecular hydrogen bond. The next band observed in the range of 3310–3230 cm^−1^ originates from the vibrations of the 6-OH···O-3′ inter-molecular hydrogen bond in cellulose [47]. An additional component at about 3530 cm^−1^ was observed for the transgenic fibres only. Its appearance suggests the formation of free additional hydroxyl groups in transgenic fibres. Moreover, the peak position of other bands corresponding to the inter-molecular HBs of transgenic samples was clearly higher than those of the control sample and, simultaneously, the integral intensities of bands originating from intra- and inter-molecular hydrogen bonds in transgenic samples were lower than those in control fibres. This proves that the number of the HBs in transgenic samples was smaller than those in natural sample but these bonds were stronger in the former case. Consequently, the molecular chains of cellulose were more flexible in the transgenic samples.

The bands observed in the range of 1500–500 cm^−1^ should be assigned to the following vibrations: at 1427 cm^−1^—δas(CH3,CH2); at 1367 cm^−1^—δs(CH3,CH2); at 1314 cm^−1^—δ(CH) + δ(OH); in the range of 1200–1300 cm^−1^—ν(C-C) and ν(C-O); at 1163 cm^−1^—δ(-OH); in the range of 1000–1110 cm^−1^—νas(C-O-C); in the range of 850–1000 cm^−1^—γ(CH); and in the range 500–720 cm^−1^ —δ(θ) vibrations. Another band observed at about 1460 cm^−1^ originates from the coupled (CH) + ν (-ring) vibration. All these bands are characteristic for cellulose, similar to the weak band at 899 cm^−1^ [48,49]. The integral intensity ratios of all the mentioned standardized bands originating from cellulose are very similar to those for the studied samples which means that the cellulose amount for the control and transgenic flax fibres was almost the same.

More information can be drawn from comparison of the intensities of the bands at 1054 and 994 cm^−1^, which originate from the ν(COC) vibrations of the α-1,4-glycosidic bonds of the cellulose chains. The ratios of integral intensities of these bands and the integral intensity of the 2920 cm^−1^ band were clearly higher for the control sample than for the transgenic samples (Figure 11c). These data suggest that the transgenic fibres contain fewer C-O-C bridges than those appearing in the control NIKE sample. It confirmed that the molecular chains of cellulose were more flexible in the transgenic samples and the length of these chains was shorter.

It has been commonly accepted that the spectral contour from the 1600–1800 cm^−1^ range of the IR spectrum characterizes the pectin content in the flax which plays a diagnostic role [46]. The ratios of integral intensities of the bands characteristic for pectin (1733, 1646, and 1600 cm^−1^) and integral intensity of the 2920 cm^−1^ band are very similar (Figure 11d) which suggest that the control and transgenic fibres contain similar amounts of pectin.

The useful information on the lignin content in the flax samples can be derived by comparing the spectral contours from the range of 1200–1550 cm^−1^. The bands at 1507, 1337, and 1260 cm^−1^ originate from vibrations of lignin constituents [47]. The ratios of their integral intensities standardized to the integral intensity of the 2920 cm^−1^ band exhibited similar amounts of lignin appearing in all the studied samples, control and transgenic (see Figure 11e).

The so-called “crystallinity index”, Icr, is an important factor that characterizes the structural ordering of cellulose macromolecules in different media [50,51]. It can be calculated from the intensity ratio of the IR bands at 1370 and 2920 cm^−1^. The following values compare these parameters for the control and transgenic samples: CTR—4.7, C10—1.9, C47—1.9. Hence, the Icr values were lower in the transgenic samples and the structural ordering was greater in control flax.

Powder XRD diffractograms of natural samples present valuable information on the properties of natural products such as flax fibres. Figure 11e shows the XRD spectra of the control (NIKE) and transgenic samples. Inspection of the observed XRD curves proved that they were very similar in the whole 10–30 2θ range. All diffraction profiles were typical for the I polymorph of cellulose with characteristic intense peaks at 22.55 and a double pattern at about 15, corresponding to the (200), (110), and (1–10) crystallographic planes, in agreement with the characteristic diffraction peaks of cellulose I_β_ [51]. The peak indexed as (200) characterizes the “unit cell” parameter of the cellulose used for the production of the tissue. Approximating this curve by a Lorentz function, its maximum is located at 22.55 for the transgenic samples but this peak appeared at 22.21 for the control flax NIKE. This means that the (200) peak is shifted for transgenic flax towards higher diffraction angles relative to the standard pattern of the control sample. This proves that the cellulose chains of transgenic flax are bonded in the (200) direction only by weak electrostatic bonds and van der Waals forces. The patterns observed at 15° for the studied samples did not show any position change but the relative intensities of the peaks indexed as (1–10) and (110) were slightly different and were smaller than those of the (200) peak.

## 4. Discussion

Our previous research showed that the expression of the bacterial β-ketothiolase (phbA) gene influences the expression of endogenous genes of the 3-HB synthesis pathway and the 3-HB content in in vitro grown transgenic flax [28]. However, as flax is an industrial plant, the effect of 3-HB accumulation on the properties of flax raw materials is more important as it will have greater influence on the uses of flax products. In some cases, introduction of foreign genes can have negative effects on plant growth and productivity as was previously observed in flax producing polyhydroxybutyrate. In previous studies, flax explants transformed with a triple construct [including β-ketothiolase, acetoacetyl-CoA reductase, and PHB synthase genes (all under the 35S promoter) involved in the synthesis of PHB] cultured in vitro showed clearly dramatic changes in plant phenotypes including growth inhibition. Replacement of the 35S promoter with the 14-3-3 gene promoter returned the plant growth to normal levels in vitro [30]. Therefore, in the current study, we examined whether the features of the transgenes remain in homozygous 3rd generation plants cultivated in a greenhouse and how overexpression of beta-ketothiolase affects the quality of retting and flax fibres. We analysed the mature stems and the fibres from two selected transgenic lines of the F3 generation (named C10 and C47), which showed a significantly higher content of 3-HB in 5-week-old plants grown in a greenhouse. In our case, no significant changes in phenotype and productivity were observed in transgenic plants with overexpression of the bacterial beta-ketothiolase gene cultivated in a greenhouse. Notably, we observed an increase in hydroxybutyrate content only during plant growth. In mature flax stems, the content of hydroxybutyrate was even lower than in control plants. However, the accumulation of polyhydroxybutyrate was noticed. It was lower than that measured in the triple construct plants but was still high enough to possibly influence fibre properties. At this stage, it is difficult to fully explain this phenomenon as polyhydroxybutyrate synthesis in plants has not been fully elucidated. It is possible that initial higher hydroxybutyrate levels may influence downstream gene expression and/or activity. Accumulation of PHB may in turn influence the deposition of other polymers.

We examined all major cell wall components like lignin, pectin, and cellulose as they affect flax retting and fibre quality. However, we also evaluated bioactive compound content as flax products can serve as a good source of biopharmaceuticals including wound dressings based on flax fibres [52]. Fibres, obtained from flax straw, are the main flax raw material, and depends fibre flax variety. In order to facilitate the separation of the fibres, a retting process is carried out. Retting of flax straw can occur in several ways, but the key to all methods is the activity of microorganisms, which allows the separation of the fibres from the woody part. The process of retting is largely influenced by the content of lignin, which is a mechanical barrier for pectin-degrading enzymes. In addition to the lignin content, the length and efficiency of retting may be affected by the amount of pectin in the straw: the less pectin, the shorter and more efficient the retting time [53,54,55]. Components of pectin have been shown to be able to trigger defence responses, including the accumulation of reactive oxygen species, and they are monitored by wall-associated kinases and have common signalling elements [56]. Paradoxically, pectin reduction has a positive impact for industry as less pectin in the stem leads to easier and quicker retting, and subsequent higher quality of the fibres.

The straw of the transgenic plants contained a lower content of pectin and lignin, which translated into an improvement in the retting process, which was more efficient in the straw of the transgenic plants compared to the straw of the control plants. Flax with PHB was also characterized by a more effective retting process, although in addition to a decrease in pectin content, a more pronounced decrease in lignin content was also observed [31,32]. The content of the main polymers of the cell walls of the fibres obtained from the straw of the tested plants with beta-ketothiolase overexpression, as well as fibres from plants with PHB overexpression, did not change significantly [32]. Research by Wróbel and co-authors showed that the synthesis of PHB affects the amount of hemicelluloses. A decrease in the level of these compounds and their precursors was observed [31]. Based on transgenic studies, the available data suggest compensatory regulation of lignin and cellulose deposition; the decrease in lignin content led to an increase in cellulose content, and hence the values of the ratio of lignin to cellulose can indicate a potential disturbance in cell wall assembly [57,58]. A slight change in the lignin-to-cellulose ratio was observed in transgenic plants; however, major differences were observed in cellulose structure as measured by infrared spectroscopy. The change in cellulose crystallinity index is an important factor that characterizes the structural ordering of cellulose macromolecules and the number of hydrogen bonds in the cellulose. The last one can be the result of incorporation of additional molecules—polyhydroxybutyrate and maybe also hydroxybutyrate—into cell walls as it was previously reported in PHB-producing plants [32]. This may in turn influence the mechanical properties of the resulting fibres as was previously observed [59] for PHB-producing plants. These parameters were not measured here due to limited amount of obtained fibres but can be expected based on the other results. PHB-containing fibres displayed improved mechanical properties so it is possible that similar results can be achieved with single gene overexpression.

In the next step, the influence of modifications on the content of phenylpropanoid compounds in raw materials was monitored because the literature shows that ketothiolases are co-expressed with various phenolic acid and flavonoid biosynthesis genes [60,61,62]. However, flax with beta-ketothiolase overexpression showed a reduction in the levels of the majority of the phenolic compounds in straw and fibres. It did not negatively affect the antioxidant potential of flax stem extracts; on the contrary, the transgenic plants were distinguished by an increase in antioxidant potential (Appendix A). Only line C47 showed a slight increase in the content of flavonoids in flax straw, while their content in fibres decreased compared to the control. This observation is in contrast to plants overproducing PHB where increased level of phenolic acid [31] was observed but larger differences were seen in flax fibres and the resulting fabric than in whole stems [4,63].

As mentioned previously, beta-ketothiolase is connected with fatty acid metabolism. Ketothiolases are co-expressed with various plant genes, such as genes involved in beta-oxidation [9,15]. In addition, flax fatty acids have been valued for centuries as beneficial to human health so any changes in fatty acid composition of seeds and fibres need to be monitored in order to evaluate the commercial suitability of the resulting cultivar. Beta-ketothiolase is an enzyme involved in the degradation of these compounds. For example, breakdown of lipids during drought stress coincided with the increased activity of acyl-CoA synthase and dehydrogenase; the latter also participates in 3-HB biosynthesis [64]. In agreement with this, transgenic flax showed a statistically significant reduction in the levels of fatty acids. This is most likely due to the fact that overexpression of beta-ketothiolase leads to increased fatty acid beta-oxidation, which reduces their total content in plants. It is also possible that the main substrate for the synthesis of fatty acids, acetyl-CoA, was partly used for the synthesis of beta-hydroxybutyrate and subsequently PHB. Similar reductions in the content of fatty acids was also observed in flax straw with overproduction of PHB, where the content of stearate (18:0), palmitate (16:0), and linoleate (18:2) was decreased [31].

A decrease in fatty acids was also observed in the straw and seeds of plants with overproduction of polyhydroxybutyrate [31]. A similar effect was particularly visible in decrease in stearic and linoleic acids in both types of transgenic plants with overproduction of beta-hydroxybutyrate and polyhydroxybutyrate. The seeds of plants with PHB also showed a downward trend in the content of fatty acids except for linoleic acid, the content of which increased in the seeds of these plants, in contrast to the seeds of plants overexpressing beta-ketothiolase, where the content of this fatty acid was also reduced [30].

The determined levels of fatty acids in the flax straw with PHB differed from those measured in the seeds. In flax seeds with overproduction of PHB, a decrease in the content of stearic acid was observed, while the levels of palmitate and linoleate acids were higher than in the control. A decrease in the content of C18:1, C18:3, and C22:0 linoleate was also observed [30]. The seeds of plants with overproduction of 3-HB were characterized by a decrease in the amount of C18:0 and C18:2 acids. In general, newly generated transgenic plants contain lower levels of oil in seeds, potentially lowering their agricultural potential. However, it needs to be noted that the modified fibre variety of flax, are cultivated for fibres and not for oil production. Even though seeds contain slightly less oil, the oil composition was mostly the same and this did not influence seed viability. In stems, lower levels of fatty acids were also observed but this did not affect plant growth or phenotypes.

The levels of fat-soluble components were also measured. In this case, transgenic flax showed a significant reduction in the main carotenoids. β-carotene and lutein levels were probably connected to significant decrease in chlorophyll content as carotenoids serve as protective agents for the photosynthetic apparatus. This may reduce the photosynthetic apparatus protection by carotenoids upon excessive ROS formation [65]; however, major effects on plant growth were not observed.

The other major terpenoid components present in plants are sterols. Sitosterol, campesterol, and stigmasterol are the most abundant sterols in flax plants. The measurement of the amount of sterols in transgenic flax overexpressing beta-ketothiolase showed no significant changes compared to control plants, again in contrast to PHB-producing plants that had higher levels of sterols in both seeds and fibres [31].

To summarize, even though the plants overexpressing just beta-ketothiolase were proven to also result in PHB accumulation, the overall effect on plant metabolism differed in many aspect. It can be speculated that the initial increase in PHB may influence the expression of genes in various pathways. In PHB-producing plants, the additional hydroxybutyrate is simultaneously converted into PHB and accumulates in chloroplasts. Here, PHB synthesis is seemingly activated only at later stages of plant growth. One of the reasons for that is a possibility that PHB synthesis in plants is activated in response to stress as was observed in bacteria. PHB synthesis has been also associated with protective mechanisms against osmotic and oxidative stresses [66,67] and drying of plants after reaching the physiological stage of maturity which can be treated as a severe stress [68].

There is also lot of evidence, generated mostly from work on animals, showing that 3-HB is not only an intermediate metabolite, but also an important regulatory molecule that can influence gene expression mostly via the inhibition of histone deacetylase activity and thus the epigenetic regulation of many genes [69]. Our earlier work indicated that hydroxybutyrate can also act as a regulatory molecule in plants [28]. The exact mechanism needs to be further investigated; however, the presence of this novel regulatory molecule in plants and its possible influence on the expression of native genes and accumulation of PHB without the need for transgenesis open new avenues for research. This is important particularly in Europe where GMO plants face many regulatory restrictions. It would be valuable if PHB synthesis could be achieved on satisfactory levels in non-transgenic plants as PHB-containing fibres have proved to be good materials, for example, in wound dressings [4]. We believe that the research presented here is the first step towards this goal.

## 5. Conclusions

The overexpression of beta-ketothiolase reduced the content of pectin and lignin in flax stems, which contributed to a more efficient retting process. In the tested flax stems, a decrease in the content of fatty acids, carotenoids, and phenolic compounds associated with the cell wall was observed, while a slight increase in free flavonoids was noted. The recorded changes in the content of the above metabolites did not adversely affect the growth and development of the plant, as well as the application properties of raw materials obtained from these plants. Flax fibres were characterized by a reduced content of phenolic compounds and changes in the structure of the cell walls. In the transgenic fibres, the molecular chains of cellulose were more flexible and the length of these chains was shorter, with smaller Icr values; thus, the structural order of the cellulose was smaller. The observed changes in the metabolism of plants with overproduction of beta-hydroxybutyrate were similar to those noted in plants with overproduction of polyhydroxybutyrate, which may indicate that some of the changes in plants with PHB are caused by its monomer. However, to confirm that beta-hydroxybutyrate or PHB can influence gene expression more research is needed. This could be done, for example, via creation and analysis of transgenic plants with various levels and/or ratios of HB and PHB and investigation on the beta-hydroxybutyrate signalling route.

## Figures and Tables

**Figure 1 metabolites-13-00437-f001:**
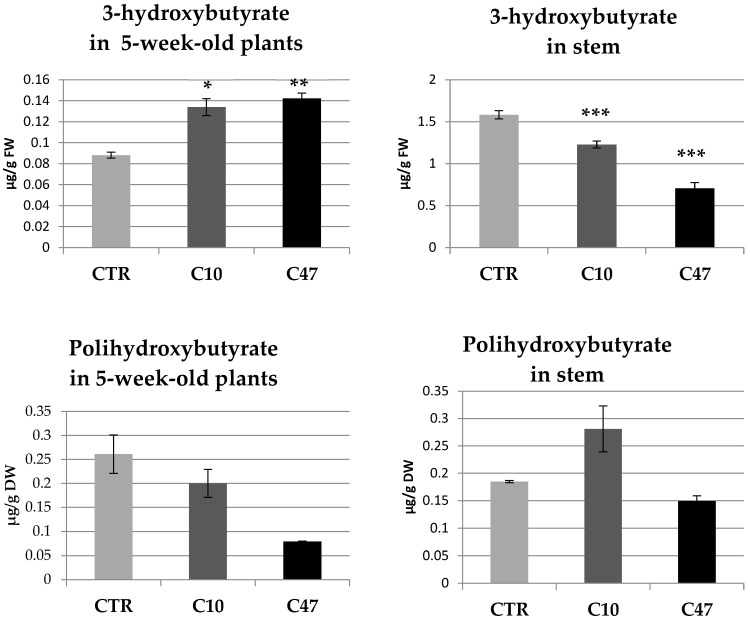
3-hydroxybutyrate and polyhydroxybutyrate content in 5-week-old plants and stems in transgenic flax lines C10 and C47 and in non-transgenic flax (CTR). The results are depicted as the mean of three biological repetitions ± SD. Asterisks indicate statistically significant changes (*—*p* < 0.05, **—*p* < 0.01, ***—*p* < 0.001).

**Figure 2 metabolites-13-00437-f002:**
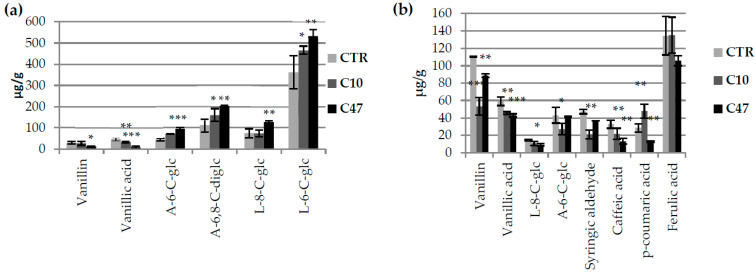
Levels of selected phenolic compounds in transgenic and control plants. Content of phenolic compounds in flax stem of maturation stage [(**a**)—free phenolic compounds: vanillin, vanillic acid, 6,8-C-diglucoside of apigenin (A-6,8-C-diglc), 6-C-glucose of luteolin (L-6-C-glc), 8-C-glucoside of luteolin (L-8-C-glc), and 6-C-glucoside of apigenin (A-6-C-glc) and (**b**)—bound phenolic compounds: vanillin, vanillic acid, syringaldehyde, luteolin 8-C-glucoside, apigenin 6-C-glucoside, p-coumaric acid, caffeic acid and ferulic acid] in transgenic flax lines C10 and C47, and control flax (CTR). The results are depicted as the mean of three biological repetitions ± SD. Asterisks indicate statistically significant changes (*—*p* < 0.05, **—*p* < 0.01, ***—*p* < 0.001).

**Figure 3 metabolites-13-00437-f003:**
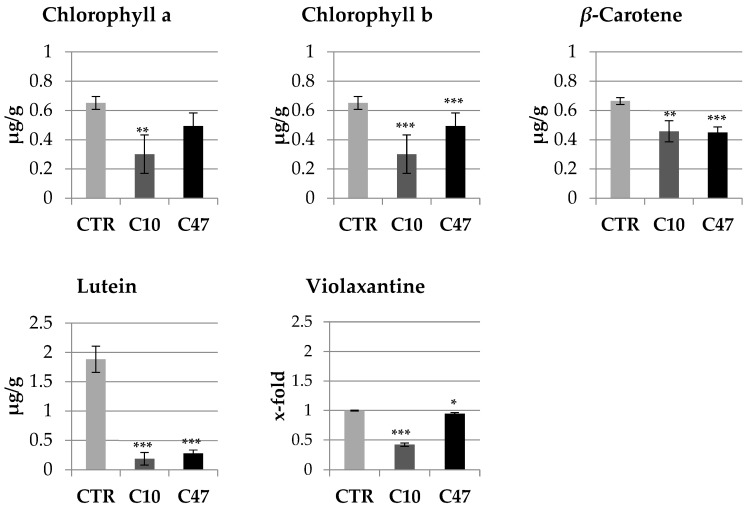
Content of terpenoids in mature stems of transgenic flax lines C10 and C47 and control plants (CTR). The contents of chlorophyll a, chlorophyll b, beta-carotene, lutein, and violaxantine were obtained from the UPLC analysis. Calculations were performed using MassLynx 2.0 (Waters, Milford, CT, USA) software. The results are depicted as the mean of three biological repetitions ± SD. Asterisks indicate statistically significant changes (*—*p* < 0.05, **—*p* < 0.01, ***—*p* < 0.001).

**Figure 4 metabolites-13-00437-f004:**
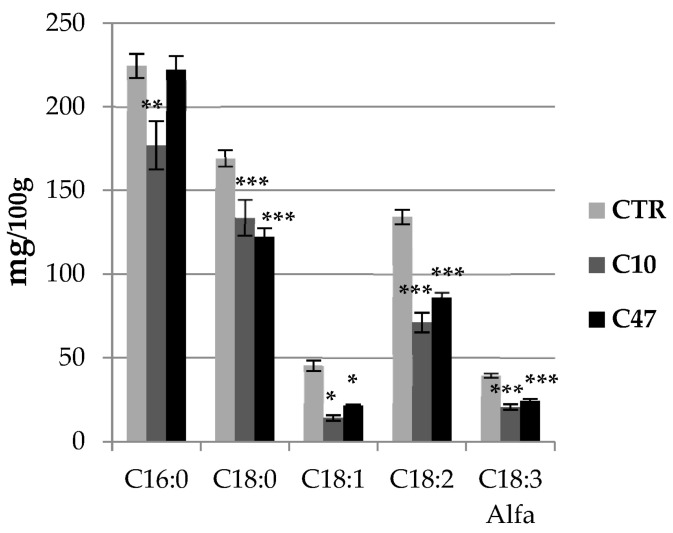
Fatty acid contents in mature stem of transgenic flax lines C10 and C47 and control plants (CTR). Fatty acid (C16:0, C18:0, C18:1, C18:2, and C18:3 alfa) contents were determined using the GC-FID method. The results are depicted as the mean of three biological repetitions ± SD. Asterisks indicate statistically significant changes (*—*p* < 0.05, **—*p* < 0.01, ***—*p* < 0.001).

**Figure 5 metabolites-13-00437-f005:**
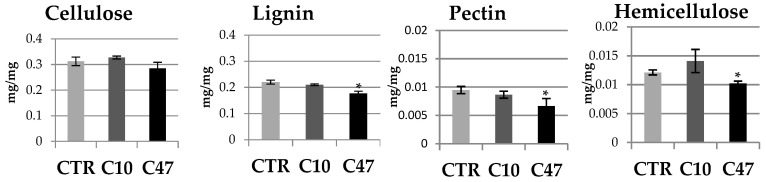
Contents of cellulose, lignin, pectin, and hemicellulose in the cell walls of mature stems of transgenic flax lines C10 and C47 and control (CTR). The results are depicted as the mean of three biological repetitions ± SD. Asterisks indicate statistically significant changes (*—*p* < 0.05).

**Figure 6 metabolites-13-00437-f006:**
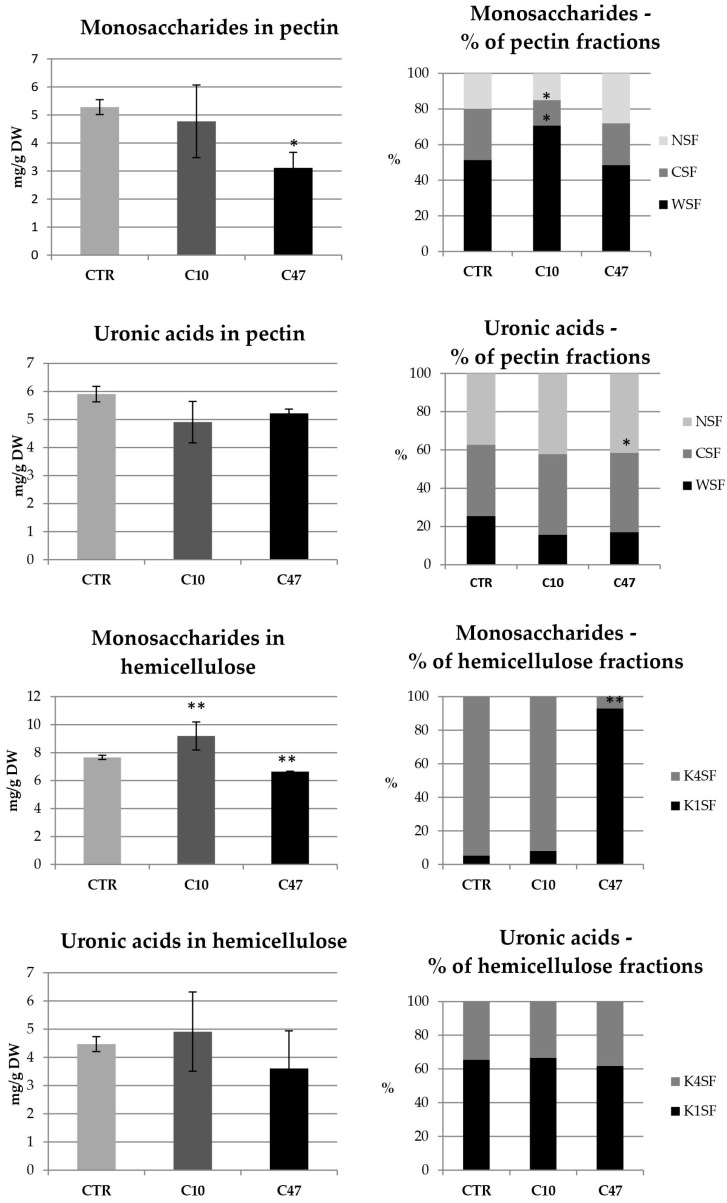
Contents of monosaccharides and uronic acids in pectin and hemicellulose as well as percentage of monosaccharides and uronic acids in pectin and hemicellulose fractions in the cell walls (WSF, CSF, NSF, K1SF, and K4SF) of mature stems of transgenic flax lines C10 and C47 and control (CTR). The results are depicted as the mean of three biological repetitions ± SD. Asterisks indicate statistically significant changes (*—*p* < 0.05, **—*p* < 0.01).

**Figure 7 metabolites-13-00437-f007:**
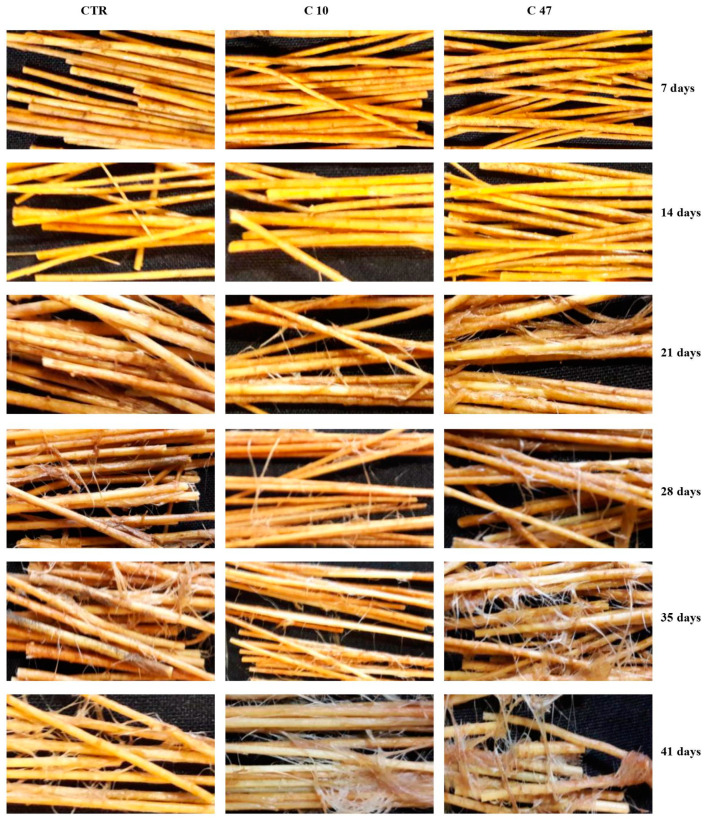
Analysis of efficiency of retting for flax transgenic plants (C10 and C47) and control plants (CTR) after 7, 14, 21, 28, 35, and 41 days.

**Figure 8 metabolites-13-00437-f008:**
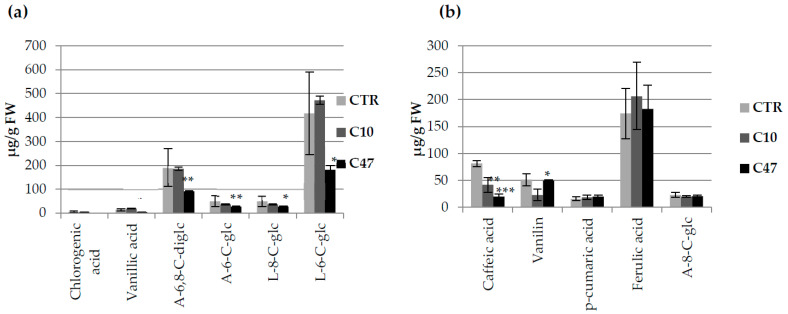
Levels of selected phenolic compounds in transgenic and control flax fibres. Content of phenolic compounds in fibres [(**a**)—free phenolic compounds: vanillic acid, chlorogenic acid, 6,8-C-diglucoside of apigenin (A-6,8-C-diglc), 6-C-glucose of luteolin (L-6-C-glc), 8-C-glucoside of luteolin (L-8-C-glc), and 6-C-glucoside of apigenin (A-6-C-glc) and (**b**)—bound phenolic compounds: vanillin, apigenin 8-C-glucoside (A-8-C-glc), p-coumaric acid, caffeic acid, and ferulic acid] from transgenic flax lines C10 and C47, and control flax (CTR). The results are depicted as the mean of three biological repetitions ± SD. Asterisks indicate statistically significant changes (*—*p* < 0.05, **—*p* < 0.01, ***—*p* < 0.001).

**Figure 9 metabolites-13-00437-f009:**
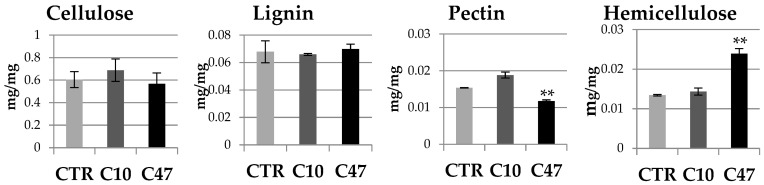
Content of cellulose, lignin, pectin, and hemicellulose in the cell wall of fibres of transgenic flax lines C10 and C47 and control (CTR). The results are depicted as the mean of three biological repetitions ± SD. Asterisks indicate statistically significant changes (**—*p* < 0.01).

**Figure 10 metabolites-13-00437-f010:**
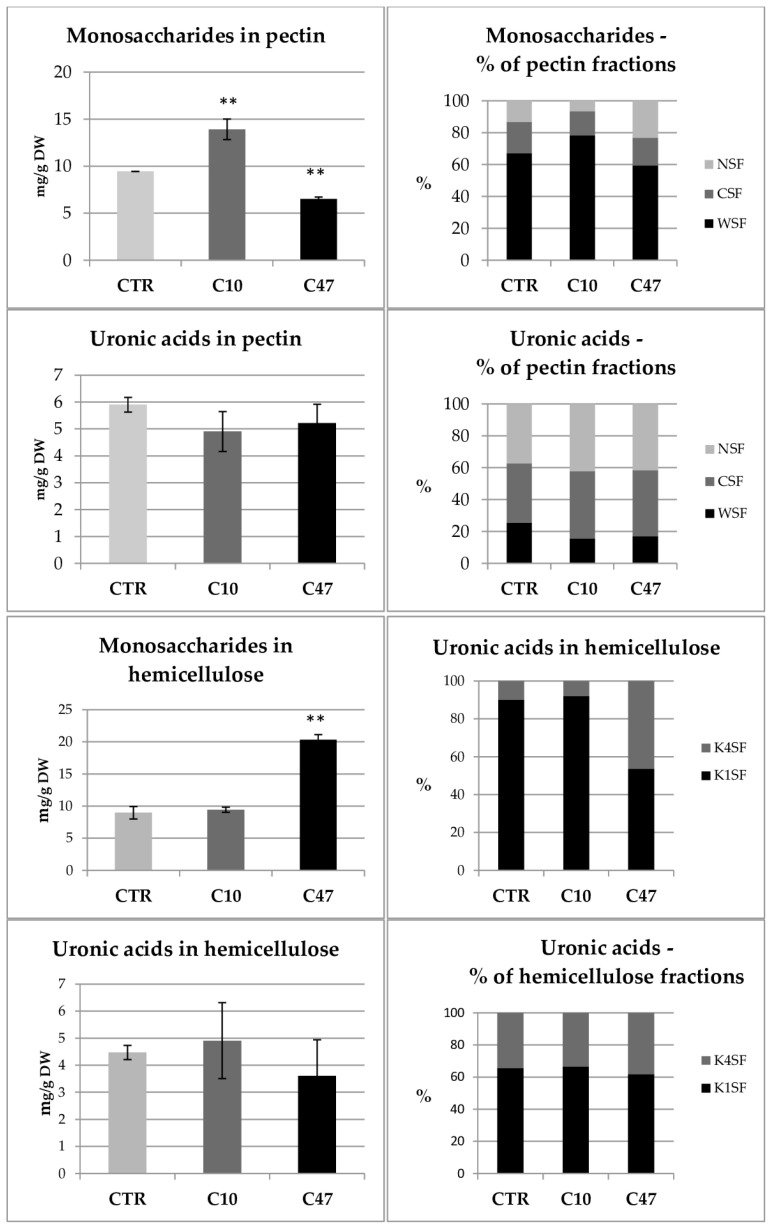
Content of monosaccharides and uronic acids in pectin and hemicellulose as well as percentage of monosaccharides and uronic acids in pectin and hemicellulose fractions in the cell walls (WSF, CSF, NSF, K1SF, and K4SF) of fibres of transgenic flax lines C10 and C47 and control (CTR). The results are depicted as the mean of three biological repetitions ± SD. Asterisks indicate statistically significant changes (**—*p* < 0.01).

**Figure 11 metabolites-13-00437-f011:**
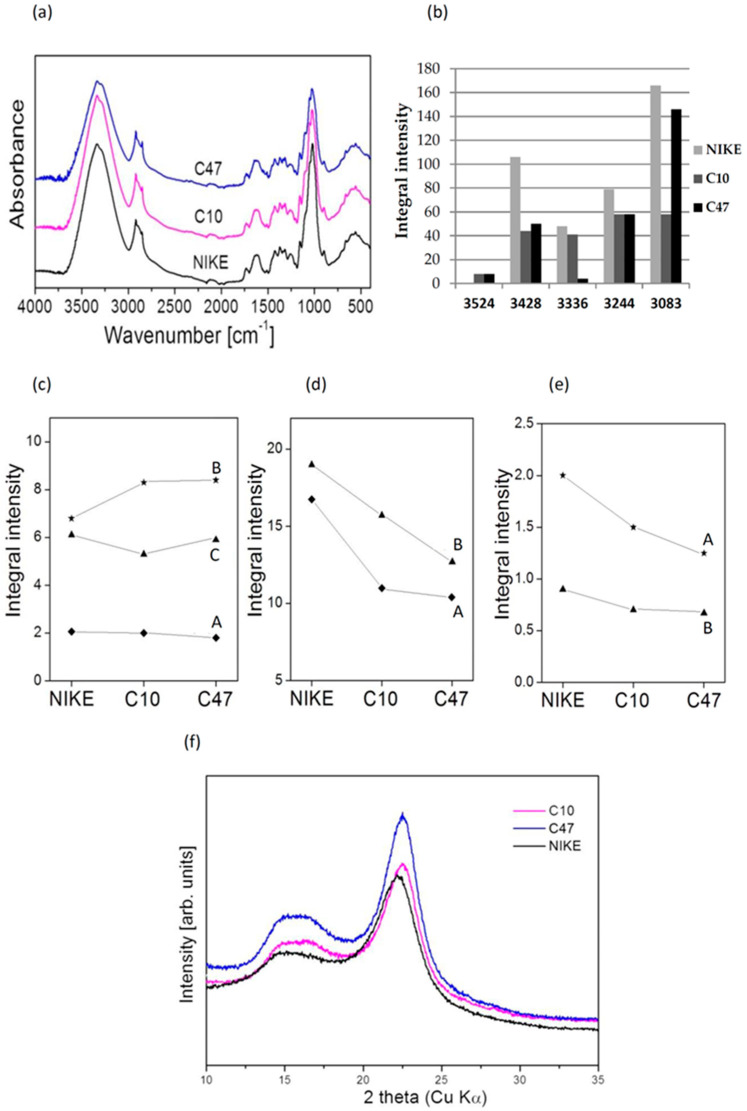
(**a**) IR spectra of fibres from transgenic flax lines C10 and C47 and non-transgenic flax Nike. (**b**) Wavenumber (ν) and integral intensity ratio of Lorentzian component related to the 2920 cm^−1^ standard (A) observed for non-transgenic (NIKE) and transgenic flax fibres. (**c**) Differences in the integral intensities of the bands at 1733 cm^−1^ (A), 1646 cm^−1^ (B), and 1600 cm^−1^ (C) for the NIKE, C10, and C47 samples. (**d**) Differences in the integral intensities of the bands at 1054 cm^−1^ (A) and 994 cm^−1^ (B) for the NIKE, C10, and C47 samples. (**e**) Differences in the integral intensities of the bands at 1507 cm^−1^ (A) and 1337 cm^−1^ (B) for the NIKE, C10, and C47 samples. (**f**) XRD diagrams of non-transgenic (NIKE) and transgenic flax samples.

## Data Availability

Data available on request. Data is not publicly available due to privacy or ethical restrictions.

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
