# Peer review of "Overexpression of Bacterial Beta-Ketothiolase Improves Flax (Linum usitatissimum L.) Retting and Changes the Fibre Properties"

_metabolites, 2023, doi:10.3390/metabo13030437_

Round 1

Reviewer 1 Report

Mierziak et al. studied the effect of overexpression of beta-ketothiolase and its regulatory changes in stem and fibre composition and hydroxybutyrate of flax plants. This topic is relevant to Metabolites, because they studied about hydroxybutyrate, poly hydroxybutyrates, phenolic compounds, terpenoids, cellulose, lignin, pectin and hemicellulose. Compared to other published articles, they estimated the most of metabolites to conclude their studies.  Manuscript was well written and easy to read but avoid two sentences as a one paragraph. Conclusion looks general, please rewrite the conclusion with specific evidence and for future research. They addressed the main question posed.

There are some comments:

Comment 1: Write the scientific name of the plant on title and abstract.

Comment 2: Write expansion for abbreviations in abstract.

Comment 3: Lot of information in the main text, please rewrite the both abstract and conclusion.

Comment 4: Avoid two sentences as a paragraph, please check flow of writing and English.

Author Response

Thank you very much for your valuable review. We tried to incorporate all your suggestions into revised manuscript. We also read carefully manuscript and rewritten abstract and conclusion. We hope that our paper was sufficiently improved for publication.

We list the answers to questions and reviewers suggestion below.

  1. Mierziak et al. studied the effect of overexpression of beta-ketothiolase and its regulatory changes in stem and fibre composition and hydroxybutyrate of flax plants. This topic is relevant to Metabolites, because they studied about hydroxybutyrate, poly hydroxybutyrates, phenolic compounds, terpenoids, cellulose, lignin, pectin and hemicellulose. Compared to other published articles, they estimated the most of metabolites to conclude their studies. Manuscript was well written and easy to read but avoid two sentences as a one paragraph. Conclusion looks general, please rewrite the conclusion with specific evidence and for future research. They addressed the main question posed.

There are some comments:

Comment 1: Write the scientific name of the plant on title and abstract.

We added scientific name of the plant on title and abstract.

Comment 2: Write expansion for abbreviations in abstract.

We replaced abbreviations with full names in the abstract

Comment 3: Lot of information in the main text, please rewrite the both abstract and conclusion.

The manuscript was read again and the both sections:  abstract and conclusion were rewritten.   

Comment 4: Avoid two sentences as a paragraph, please check flow of writing and English.

The layout of paragraphs was changed and the manuscript was read again and corrected when necessary

Reviewer 2 Report

The manuscript submitted for review presents a well-designed and properly conducted study that may focus the attention of Metabolites readers. My minor comments relate to the description of UPLC methods. In my opinion, the authors, in addition to citing previous publications that described the study procedures, should indicate the type of detection - MS, DAD, or other. As they did for the GC-FID method. In Figures 2 and 8, the descriptions of flavonoids have been clipped, which disrupts the presentation. I suggest using acronyms with appropriate descriptions (e.g., L-8-C-glc, L-6,8-diglc, A-6-C-glc) or the following scheme for recording the compound name: luteolin 8-C-glucoside instead of 8-C-glucoside of luteolin. I also ask that vanillin and vanillic acid be reclassified from "benzoic compounds" to "benzene-based compounds," since vanillin is an aldehyde (4-hydroxy-3-methoxybenzaldehyde), not a carboxylic acid.

Author Response

Thank you very much for your valuable review. We tried to incorporate all your suggestions into revised manuscript. We also read carefully manuscript and rewritten abstract and conclusion. We hope that our paper was sufficiently improved for publication.

We list the answers to questions and reviewers suggestion below.

The manuscript submitted for review presents a well-designed and properly conducted study that may focus the attention of Metabolites readers. My minor comments relate to the description of UPLC methods. In my opinion, the authors, in addition to citing previous publications that described the study procedures, should indicate the type of detection - MS, DAD, or other. As they did for the GC-FID method.

The description of methods was expanded including type of apparatus and detector used in the study.

In Figures 2 and 8, the descriptions of flavonoids have been clipped, which disrupts the presentation. I suggest using acronyms with appropriate descriptions (e.g., L-8-C-glc, L-6,8-diglc, A-6-C-glc) or the following scheme for recording the compound name: luteolin 8-C-glucoside instead of 8-C-glucoside of luteolin.

Figure 2 and Figure 8 were revised and acronyms with appropriate descriptions (e.g., L-8-C-glc, L-6,8-diglc, A-6-C-glc) were added in the figure legend to avoid clipped of the description.

I also ask that vanillin and vanillic acid be reclassified from "benzoic compounds" to "benzene-based compounds," since vanillin is an aldehyde (4-hydroxy-3-methoxybenzaldehyde), not a carboxylic acid.

It was changed according to reviewer recommendation.